# UNDERSTANDING ADDITION IN TRANSFORMERS

**Philip Quirke**
Apart Research

**Fazl Barez**
Apart Research
University of Oxford

## ABSTRACT

Understanding the inner workings of machine learning models like Transformers is vital for their safe and ethical use. This paper provides a comprehensive analysis of a one-layer Transformer model trained to perform $n$-digit integer addition. Our findings suggest that the model dissects the task into parallel streams dedicated to individual digits, employing varied algorithms tailored to different positions within the digits. Furthermore, we identify a rare scenario characterized by high loss, which we explain. By thoroughly elucidating the model's algorithm, we provide new insights into its functioning. These findings are validated through rigorous testing and mathematical modeling, thereby contributing to the broader fields of model understanding and interpretability. Our approach opens the door for analyzing more complex tasks and multi-layer Transformer models.

## 1 INTRODUCTION

Understanding the underlying mechanisms of machine learning models is essential for ensuring their safety and reliability (Barez et al., 2023; Olah et al., 2020b; Doshi-Velez and Kim, 2017; Hendrycks and Mazeika, 2022). By unraveling the inner workings of these models, we can better understand their strengths, limitations, and potential failure modes, enabling us to develop more robust and trustworthy systems. Specifically, the sub-field of *mechanistic interpretability* within machine learning interpretability aims to dissect the behavior of individual neurons and their interconnections in neural networks (Räuker et al., 2022). Recent interpretability work has explored how transformers make predictions (Neo et al., 2024), analyzed reward model divergence in large language models (Marks et al., 2024), and highlighted the importance of such analyses for measuring value alignment (Barez and Torr, 2023). This pursuit is part of a larger endeavor to make the decision-making processes of complex machine learning models transparent and understandable.

Although models like Transformers have shown remarkable performance on a myriad of tasks, their complexity makes them challenging to interpret. Their multi-layered architecture and numerous parameters make it difficult to comprehend how they derive specific outputs (Vig, 2019). Further, while simple arithmetic tasks like integer addition may be trivial for humans, understanding how a machine learning model like a Transformer performs such an operation is far from straightforward (Liu and Low, 2023).

In this work, we offer an in-depth analysis of a one-layer Transformer model performing $n$-digit integer addition. We show that the model separates the addition task into independent digit-specific streams of work, which are computed in parallel. Different algorithms are employed for predicting the first, middle, and last digits of the answer. The model's behavior is influenced by the compact nature of the task and the specific format in which the question is presented. Despite having the opportunity to begin calculations early, the model actually starts later. The calculations are performed in a time-dense manner, enabling the model to add two 5-digit numbers to produce a 6-digit answer in just 6 steps (See Fig. 1). A rare use case with high loss was predicted by analysis and proved to exist via experimentation. Our findings shed light on understanding and interpreting transformers. These insights may also have implications for AI safety and alignment.

Our results demonstrate the transformer's unique approach applies to integer addition across various digit lengths (see Appendixes B and C). This transformer architecture, with its self-attention mechanism and ability to capture long-range dependencies, offers a powerful and flexible framework for

---

Corresponding author: `fazl@robots.ox.ac.uk`.

modeling sequential data. Our theoretical framework provides a mathematical justification for the model's behavior, substantiating our empirical observations and offering a foundation for future work in this domain.

Our main **contributions** are four-fold:

- Reformulation of the traditional mathematical rules of addition into a framework more applicable to Transformers.
- Detailed explanation of the model's (low loss) implementation of the addition algorithm, including the problem and model constraints that informed the algorithm design.
- Identification of a rare use case where the model is not safe to use (has high loss), and explanation of the root cause.
- Demonstration of a successful approach to elucidating a model algorithm via rigorous analysis from first principles, detailed investigation of model training and prediction behaviours, with targeted experimentation, leading to deep understanding of the model.

Below, we provide an overview of related work (§3), discuss our methodology (§4), describe our mathematical framework (§5), our analysis of model training (§6) and model predictions (§7). We conclude with a summary of our findings and directions for future research (§9).

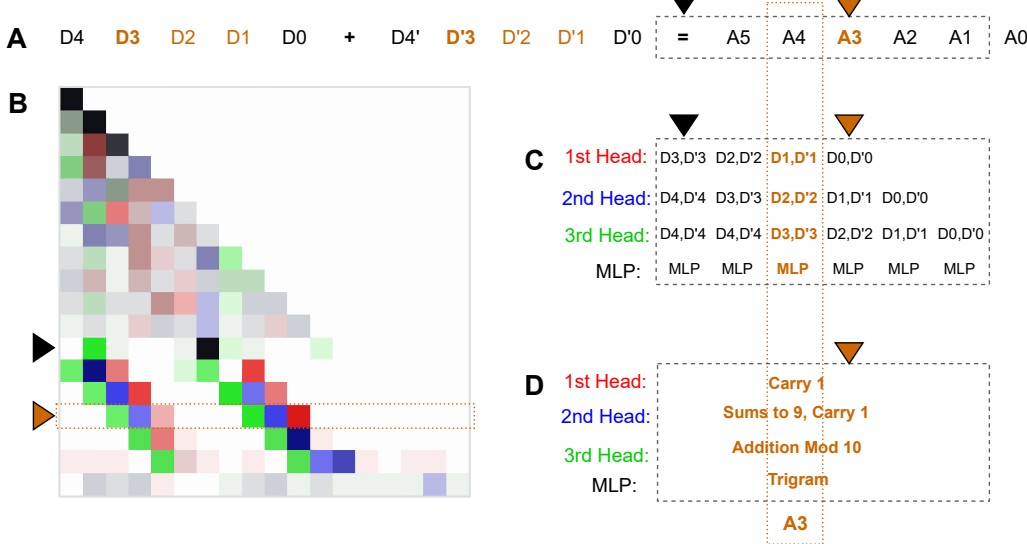

Figure 1: Illustration of the transformer model's attention pattern when adding two 5-digit integers. The model attends to digit pairs sequentially from left to right, resulting in a "double staircase" pattern across rows. **A:** The 5 digit question is revealed token by token. The "10s of thousands" digit is revealed first. **B:** From the "=" token, the model attention heads focus on successive pairs of digits, giving a "double staircase" attention pattern. **C:** The 3 heads are time-offset from each other by 1 token such that, in each row, data from 3 tokens is available. **D:** To calculate A3, the 3 heads do independent simple calculations on D3, D2 and D1. The results are combined by the MLP layer using trigrams. A3 is calculated one token before it is needed. This approach applies to all answer digits, with the first and last digits using slight variations of the approach.

## 2 BACKGROUND

We focus on a single-layer transformer model with a vocabulary of size $V$ containing a set of symbols $\mathcal{V}$. The model converts input (e.g., a sequence of symbols from $\mathcal{V}$) into an input sequence $(\mathbf{x}_1, \dots, \mathbf{x}_p)$, where each $\mathbf{x}_i \in \mathbb{R}^V$ is a one-hot vector representing the corresponding symbol from $\mathcal{V}$. The input tokens are mapped to $d_e$-dimensional embeddings by multiplying with an embedding

matrix $\mathbf{E} \in \mathbb{R}^{d_e \times V}$, where the $i$-th column of $\mathbf{E}$ represents the embedding of the $i$-th symbol in $\mathcal{V}$. The resulting sequence of embeddings is denoted as $(\mathbf{e}_1, \ldots, \mathbf{e}_p)$, where $\mathbf{e}_i = \mathbf{E}\mathbf{x}_i \in \mathbb{R}^{d_e}$. The model processes the input embeddings using a mechanism called "self-attention". Each input embedding $\mathbf{e}_i$ is passed through a self-attention mechanism that calculates weighted relationships between all input embeddings – capturing the importance of each embedding relative to others. The model then aggregates these weighted representations to produce contextually enriched representations for each embedding. The contextually enriched representations produced by the self-attention mechanism are subsequently fed through feedforward neural networks (i.e., multilayer perceptrons, MLPs) to refine their information. Finally, the output tokens are generated based on the refined representations and converted back to human-readable format using the vocabulary $\mathcal{V}$.

## 3   RELATED WORK

Interpreting and reverse engineering neural networks and transformers to find meaningful circuits has been an area of active research. Olah et al. (2020a) argued that by studying the connections between neurons and their weights, we can find meaningful algorithms (aka Circuits) in a "vision" neural network. Elhage et al. (2021) extended this approach to transformers, conceptualizing their operation in a mathematical framework that allows significant understanding of how transformer operate internally. Various tools (Foote et al., 2023; Conmy et al., 2023b; Garde et al., 2023) use this framework to semi-automate some aspects of reverse engineering. Nanda et al. (2023) reverse-engineered modular addition (e.g. 5 + 7 mod 10 = 2) showing the model used discrete Fourier transforms and trigonometric identities to convert modular addition to rotation about a circle.

Nanda and Lieberum (2022) have argued models comprise multiple circuits. They gave examples, including the distinct training loss curve per answer digit in *5-digit* integer addition, but did not identify the underlying circuits. This work investigates and explains the circuits in **n-digit** integer addition.

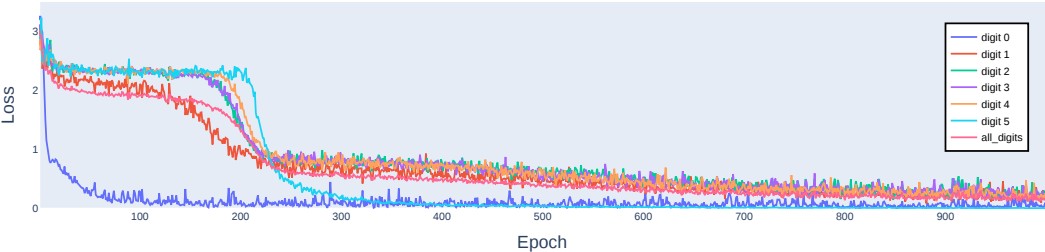

Figure 2: For 5-digit integer addition, these per-digit training loss curves show the model trains each answer digit semi-independently. Answer digit 0 (the "units" digit) is learnt much more quickly than other digits.

Circuit analysis can extract graphical circuit representations and analyze component interactions (Bau et al., 2017). To enable analysis and interpretability, techniques in works like Petersen et al. (2021) symbolically reverse-engineer networks by recovering computational expressions. Research including Seth (2005) advocate analyzing networks causally, introducing structural causal models to infer mechanisms. Examinations of sequence models like Petersen et al. (2021) have analyzed the emergence and interaction of modular components during training. Lan and Barez (2024) analyzed and compared shared circuit subgraphs for related sequence continuation tasks in a large language model. Evolutionary perspectives such as Miikkulainen (2021) elucidate how selection pressures shape hierarchical representations. Lo et al. (2024) show how such representations, even if pruned, can quickly re-emerge in models after little re-training.

Information bottleneck analyses including Kawaguchi et al. (2023) relate bottlenecks to abstractions and modularization arising from forced compression. Surveys like Carbonneau et al. (2022) overview techniques to disentangle explanatory factors into separate latent dimensions. Novel objectives

proposed in works like Conmy et al. (2023a) improve interpretability by encouraging modularity and disentanglement.

## 4 METHODOLOGY

A successful integer addition model must cope with a very large question and answer space. For 5 digit addition, there are 10 billion distinct questions (e.g. "54321+77779=") and 200,000 possible answers. Just one token (the "=") after seeing the complete question, the model must predict the first answer digit. It must correctly predict all 6 answers digits.

Figure 3: We refer to individual tokens in a 5-digit addition question as D4, .. D0, and D'4, .., D'0 and the answer tokens as A5, .., A0.

Our model was trained on 1.8 million out of 10 billion questions. After training, the model predicts answers to questions with low loss, showing the model does not rely on memorisation of training data. Fig. 2 shows the model trains each digit semi-independently suggesting the model performs integer addition by breaking down the task into parallel digit-specific streams of computation.

The Transformer model algorithms often differ significantly from our initial expectations. The easiest addition process for humans, given the digits can be processed in any order, is to add the unit digits first before moving on to higher value digits. This autoregressive transformer model processes text from left to right, so the model predicts the higher value digits (e.g. thousands) of the answer before the lower value digits (e.g. units). It can't use the human process.

A key component of addition is the need to sum each digit in the first number with the corresponding digit in the second number. Transformer models contain "attention heads",the only computational sub-component of a model that can move information *between* positions (aka digits or tokens). Visualising which token(s) each attention head focused on in each row of the calculation provided insights. While our model works with 2, 3 or 4 attention heads, 3 attention heads give the most easily interpreted attention patterns. Fig. 4 shows the attention pattern for a single 5 digit addition calculation using 3 attention heads. Appendix C shows the same pattern for 10 and 15 digit addition. Appendix C shows the pattern with 2 or 4 attention heads.

While it's clear the model is calculating answer digits from highest value to lowest value, using the attention heads, it's not clear what calculation each attention head is doing, or how the attention heads are composed together to perform addition.

## 5 MATHEMATICAL FRAMEWORK

To help investigate, we created a mathematical framework describing what **any** algorithm must do if it is to perform addition correctly. Our intuition is that the model a) incrementally discovers a necessary and sufficient set of addition sub-tasks (minimising complexity), b) discovers these sub-tasks semi-independently (maximising parallelism), and c) treats each digit semi-independently (more parallelism). Our framework reflects this.

To explain the framework, Let $D = (d_{n-1}, \ldots, d_1, d_0)$ and $D' = (d'_{n-1}, \ldots, d'_1, d'_0)$ be two $n$-digitof digits. We assert that the framework utilizes three base functions that operate on individual digit pairs. The first is **Base Add** ( aka `BA` ), which calculates the sum of two digits $D_i$ and $D'_i$ modulo 10, ignoring any carry over from previous columns. The second is **Make Carry 1** ( aka `MC1` ), which evaluates if adding digits $D_i$ and $D'_i$ results in a carry over of 1 to the next column. The third is **Make Sum 9** ( aka `MS9` ), which evaluates if $D_i + D'_i = 9$ exactly.

In addition, the framework uses two compound functions that chain operations across digits. The first is **Use Carry 1** ( aka `UC1` ), which takes the previous column's carry output and adds it to the sum of

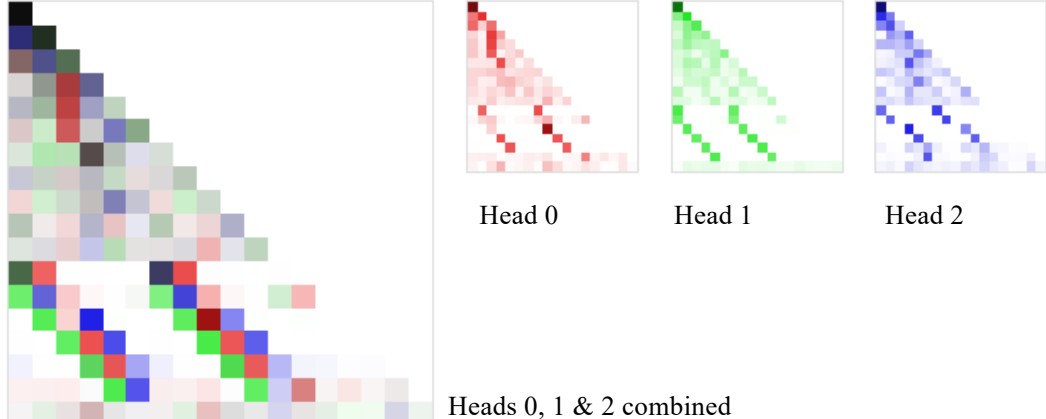

Head 0          Head 1          Head 2

Heads 0, 1 & 2 combined

Figure 4: The attention pattern, for a model with 3 attention heads, performing a single 5 digit addition. The pattern is 18 by 18 squares (as 54321+77779=132100 is 18 tokens). Time proceeds vertically downwards, with one additional token being revealed horizontally at each row, giving the overall triangle shape. After the question is fully revealed (at row 11), each head starts attending to pairs of question digits from left to right (i.e. high-value digits before lower-value digits) giving the "double staircase" shape. The three heads attend to a given digit pair in three different rows, giving a time ordering of heads.

the current digit pair. The second is **Use Sum 9** ( aka **US9** ), which propagates (aka cascades) a carry over of 1 to the next column if the current column sums to 9 and the previous column generated a carry over. **US9** is the most complex task as it spans three digits. For some rare questions (e.g. 00555 + 00445 = 01000) **US9** applies to up to four sequential digits, causing a chain effect, with the **MC1** cascading through multiple digits. This cascade requires a time ordering of the **US9** calculations from lower to higher digits.

These tasks occur in the training data with different, predictable frequencies (e.g. **BA** is common, **US9** is rarer). Compound tasks are reliant on the base tasks and so discovered later in training. The discovery of each task reduces the model loss by a different, predictable amount (e.g. **BA** by 50%, **US9** by 5%). Combining these facts give an expected order of task discovery during training as shown in Fig. 5. We use this mathematical framework solely for analysis to gain insights. The model training and all loss calculations are completely independent of this mathematical framework.

## 6 TRAINING ANALYSIS

Fig. 2 shows the model trains each digit semi-independently. Armed with the mathematical framework, we investigated each digit separately. The Digit 0 calculation is the least interesting as it only uses **BA** (not **UC1** or **US9** ). Once discovered, Digit 0 always quickly refines to have the lowest loss and least noise (as expected). (Graphs in Appendix B.)

For the other digits, we categorised the training data into 3 non-overlapping subsets aligned to the **BA,** **UC1** and **US9** tasks, and graphed various combinations, finding interesting results. The **US9** graphs are much noisier than other graphs (Fig. 6). We found that the model has low loss on simple **US9** cases (e.g. 45 + 55 = 100) but has high loss on **US9** cascades (e.g. 445 + 555 = 1000) where the **MC1** must be propagated "right to left" two 3 or 4 columns. The model can't perform these rare use cases safely, as it has a "left to right" algorithm.

Graphing the **BA** and **UC1** use cases side by side for any one of the Digits 1, 2 and 3 shows an interesting pattern (Fig. 7). In Phase 1, both tasks have the same (high) loss. In Phase 2, both curves drop quickly but the **BA** curve drops faster than the **UC1** curve. This "time lag" matches our expectation that the **BA** task must be accurate before the **UC1** task can be accurate. In Phase 3, both tasks' loss curve decrease slowly over time.

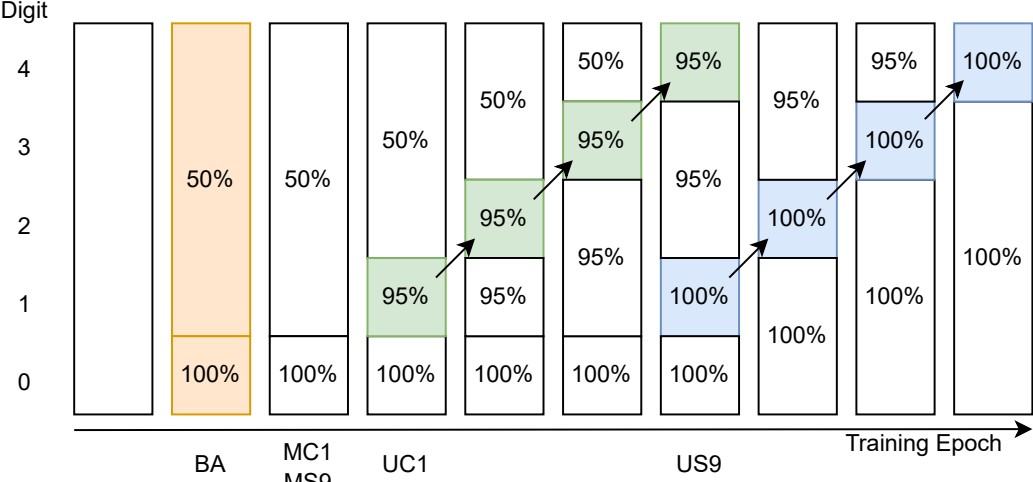

Figure 5: The mathematical framework (our method) predicts that during training, tasks are learnt for each digit independently, progressively increasing per digit accuracy (i.e. decreasing loss) shown as percentages. Mathematical rules cause dependencies between digits, giving an predicted ordering for perfect (i.e. zero loss) addition. The chain of blue squares relate to questions like 99999 + 00001 = 100000 where the **MC1** in digit 0 causes **US9** cascades through multiple other digits.

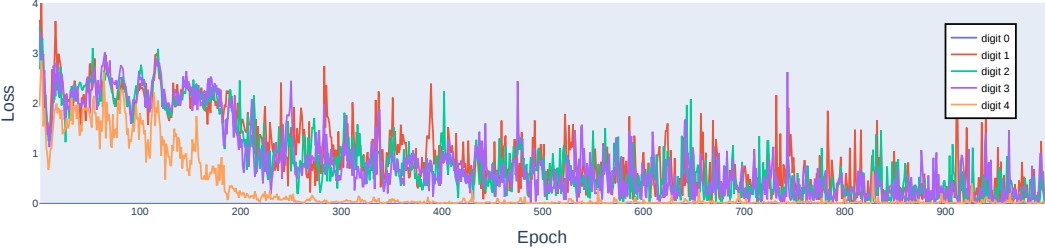

Figure 6: High variability in the per digit training loss for **US9** cases caused by the model's inability to reliably do cascading **US9** cases such as 445+555=1000.

Both the **BA** and **UC1** tasks need to move data between tokens, and so will be implemented in attention head(s). Fig. 7 shows they are trained semi-independently. We choose the number of attention heads in our model with the clearest separation of tasks in the attention pattern. We find (later) that our model has separate attention heads for the **BA** and **UC1** tasks. Digit 4, the highest question digit, has a significantly different loss curve (shown in Fig. 8) than Digits 1, 2 and 3. This is partially explained by Digit 4 only having simple use cases (i.e. no **US9** cascades). This does not explain the **BA** or **UC1** differences. This difference persists with different seed values, and with 10 or 15 digit addition. We explain this difference later.

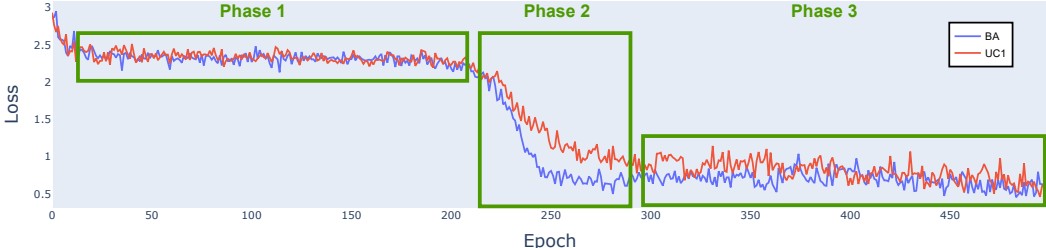

Figure 7: Training loss for digit 3 showing that, in Phase 2, the refining of **Use Carry 1** lags behind **Base Add** . **Base Add** and **Use Carry 1** are refined separately and have separate calculation algorithms. The 3 phases seem to correspond to "memorisation", "algorithm discovery" and "cleanup".

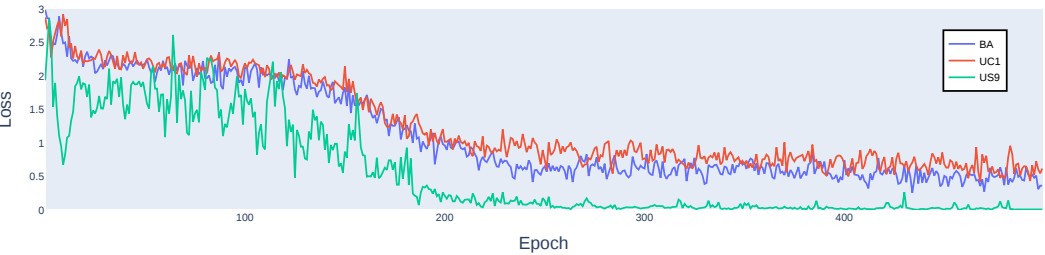

Figure 8: Training loss for digit 4 starts and stays lower for all tasks than it does for digits 1, 2 and 3. Digit 4 has a different calculation algorithm from digits 1, 2 and 3.

## 7 PREDICTION ANALYSIS

Using the ablation interventions technique we overrode (mean ablated) the model memory (residual stream) at each row (aka token position) and confirmed that the addition algorithm does **not** use any data generated in rows 0 to 10 inclusive. In these rows the model has **not** yet seen the full question and every digit in the question is independent of every other digit, making accurate answer prediction infeasible. The model also does not use the last (17th) row. Therefore, the addition answer calculation is started and completed in 6 rows (11 to 16). Further ablation experiments confirmed that the A0 to A4 answers are calculated one row before being revealed. (Details in Appendix H.)

The model has slightly different algorithms for the first answer digits (A5 and A4), the middle answer digits (A3 and A2) and the last answer digits (A1 and A0). Fig. 1 has a simplified version of how the model calculates the answer digit A3. Fig. 9 has more details. For 5 digit addition, there are 2 middle answer digits (A3 and A2) whereas for 15 digit addition there are 12 middle answer digits that use this algorithm.

The A3 addition algorithm has three clauses related to digits 3, 2 and 1. Ablating each head in turn shows that the 3rd head has most impact on loss, the 2nd head has less impact, and the 1st head has little impact. This aligns with the intuition that the sum "D3 + D'3" matters most, the `MC1` from the previous digit (D2 + D'2) matters less, and the rare `MC1` from the previous previous digit (D1 + D'1) matters least. The last two answer digits, A1 and A0, use a simplified a version of the A3 algorithm, as some clauses are not necessary.

The A3 algorithm can also be applied to A4. But the Digit 4 training curve is better (faster) than the middle digits. The attention patterns show that for A4, the model is using all the heads in row 11 (the "=" input token) when the A3 algorithm doesn't require this. Uniquely, A4 utilises more "compute" than is available to A3, A2, A1 or A0. We assume the model uses this advantage to implement a faster-training and lower-loss algorithm for A5 and A4. We haven't worked out the details of this.

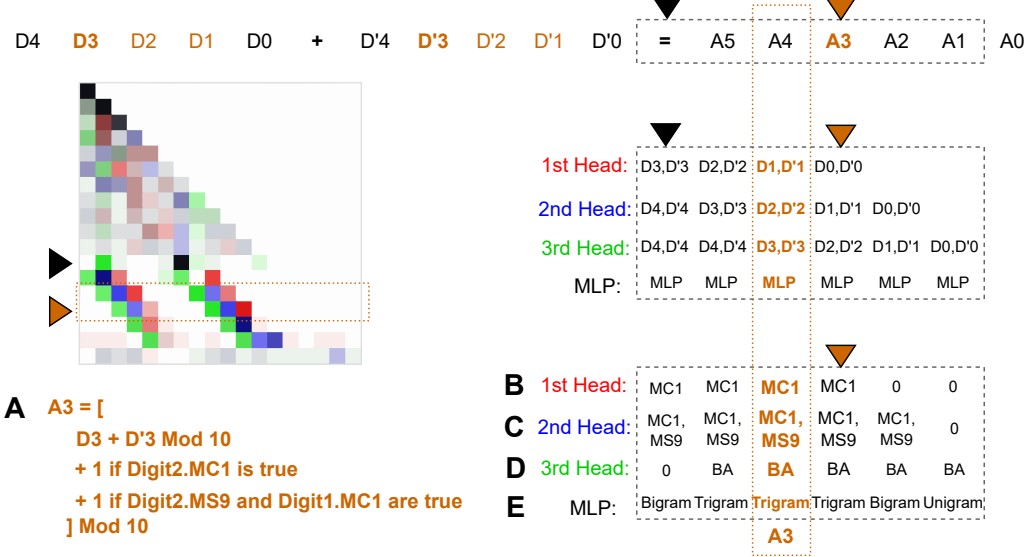

Figure 9: **A:** To predict answer digit A3, the addition algorithm must combine information from digits 3, 2 and 1. **B:** The 1st head calculates `MC1` on digit 1. **C:** The 2nd head calculates `MC1` and `MS9` (at most one of which can be true) on digit 2. **D:** The 3rd head calculates **Base Add** on digit 3. **E:** The MLP layer uses trigrams to combine the information from the 3 heads to give the final answer A3, one row before it is output. Appendix G shows this algorithm as pseudocode.

Mean ablating the 1st or 2nd head slightly increased the average loss for `BA` questions from 0.05 to 0.08, whereas ablating the 3rd head substantially increased the loss to 3.7, confirming that the 3rd head is doing the `BA` task. (Details in Appendix H.)

The MLP can be thought of as a "key-value pair" memory (Meng et al., 2022; Geva et al., 2021) that can hold many bigrams and trigrams. We claim our MLP pulls together the two-state 1st head result, the tri-state 2nd head result and the ten-state 3rd head result value, treating them as a trigram with 60 (2 x 3 x 10) possible keys. For each digit, the MLP has memorised the mapping of these 60 keys to the 60 correct digit answers (0 to 9). We haven't proven this experimentally. Our MLP is sufficiently large to store this many mappings with zero interference between mappings (Elhage et al., 2022).

Despite being feasible, the model does **not** calculate the task `MC1` in rows 7 to 11. Instead, it completes each answer digit calculation in 1 row, possibly because there are training optimisation benefits in generating a "compact" algorithm.

This algorithm explains all the observed prediction behaviour - including the fact that the model can calculate a simple `US9` case but not a cascading `US9` case. We assume that, given the dense nature of the question and answer, and the small model size, the model does not have sufficient time and compute resources to implement both `UC1` and `US9` accurately, and so preferences implementing the more common ( `UC1` ) case, and only partially implements the more complex and rare ( `US9` ) case.

## 8 ALGORITHM REUSE

We explored whether the above algorithm is learned by similar models. We trained a separate 1-layer model for 5-digit addition with a different random seed and optimization algorithm. The new model's answer format requires it to predict "+" as the first answer token, e.g., "12345+67890=+080235". Despite these changes, Figures 24, 25, and 26 show that the new model's behavior has many similarities to the previous model.

Intervention ablation demonstrates that the new model uses the BaseAdd, MakeCarry, and MakeSum sub-tasks in the same way as the previous model. The new model exhibits the same

strengths and weaknesses, e.g., it can calculate a simple MS case but not a cascading MS case. We claim that the new and previous models use essentially the same algorithm, with minor variations.

Our analysis suggests that the transformer architecture, when trained on the addition task, converges to a consistent algorithm for performing digit-wise addition. This algorithm leverages the self-attention mechanism to discover and execute the necessary sub-tasks in a parallel and semi-independent manner. Despite differences in random initialization, optimization algorithms, and output formatting, the models exhibit similar internal behavior and capabilities, indicating a robust algorithmic solution emerges from the transformer's architecture and training process.

## 9 CONCLUSIONS

This work demonstrates a successful approach to reverse engineering and elucidating the emergent algorithm within a transformer model trained on integer addition. By combining mathematical analysis, empirical investigation of training and prediction, and targeted experimentation, we are able to explain how the model divides the task into parallel digit-specific streams, employs distinct subroutines for different digit positions, postpones calculations until the last possible moment yet executes them rapidly, and struggles with a specific rare case.

Our theoretical framework of necessary addition subtasks provides a foundation for the model's behavior. The digit-wise training loss curves reveal independent refinement consistent with separate digit-specific circuits. Attention patterns illustrate staging and time-ordering of operations. Controlled ablation experiments validate our hypothesis about algorithmic elements' roles. Together these methods enable a detailed accounting of the model's addition procedure.

This methodology for mechanistic interpretability, when applied to broader tasks and larger models, can offer insights into not just what computations occur inside complex neural networks, but how and why those computations arise. Such elucidation will be increasingly important for ensuring the safety, reliability and transparency of AI systems.

## 10 LIMITATIONS AND FUTURE WORK

One concrete limitation of our current model is its difficulty handling the rare case of adding two 9-digit numbers that sum to 1 billion (e.g. 999,999,999 + 1). Further investigation is needed to understand why this specific edge case poses a challenge and to develop strategies to improve the model's performance, such as targeted data augmentation or architectural modifications.

Regarding the MLP component, a detailed ablation study could elucidate its precise role and contributions. Systematically removing or retraining this component while monitoring performance changes could shed light on whether it is essential for the overall algorithm or plays a more auxiliary part. For future work, a natural next step is to apply our framework to reverse-engineer integer subtraction models. Subtraction shares some commonalities with addition but also introduces new complexes like borrowing. Extending our approach to handle such nuances would demonstrate its generalisability.

For multiplication, an interesting avenue is to first pre-train a large transformer model on addition using our verified modules as a starting component. Then, expand the model's capacity and fine-tune on multiplication data. This could facilitate more rapid acquisition of the multiplication algorithm by providing a strong inductive bias grounded in robust addition skills. Specific experiments could then identify the emergence of new multiplication-specific subroutines and their integration with the addition circuits.

Furthermore, domains like symbolic AI, program synthesis, or general reasoning may benefit from embedding multiple specialized algorithmic components like our addition circuit within larger language models. Insights from our work could guide the controlled emergence and combination of diverse task-specific capabilities. Overall, while making models more interpretable is valuable, the ultimate aim is developing safer, more reliable, and more controllable AI systems. Our work highlight one path toward that goal through understanding of addition in neural network computations.

## 11 Reproducibility Statement

To facilitate the reproduction of our empirical results on understanding and interpreting addition in one-layer transformers, and further studying the properties of more complex transformers on more complex tasks that would build on a single layer, we release all our code and resources used in this work. Furthermore, we offer explicit constructions of one layer transformers used in this paper.

## 12 Acknowledgements

We are thankful to Jason Hoelscher-Obermaier for his comments on the earlier draft, Esben Kran for organising the Interpretability Hackathon and Clement Neo for his comments on the paper and assistance with some of the figures. This project was supported by Apart Lab.

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

## A    APPENDIX - NUMBER OF ATTENTION HEADS

The model can be successfully trained with 2, 3 or 4 attention heads (Refer Figs. 10, 11, 12). However as described in Fig. 11, the 3 attention heads case is easier to interpret than the 2 or 4 heads cases.

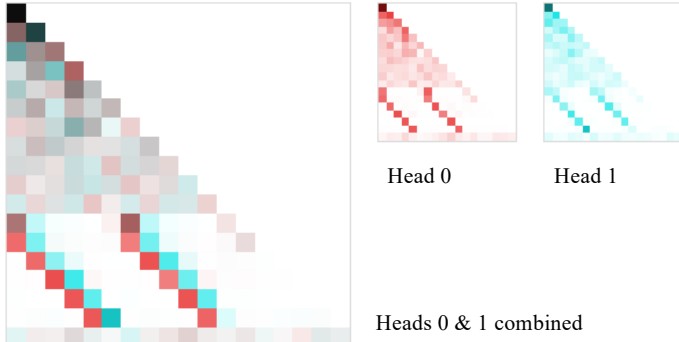

Figure 10: For 5 digit addition, using 2 attention heads works, but the model attends to multiple token pairs in a single head, suggesting that multiple tasks are being packed into a single head, which makes it harder to interpret.

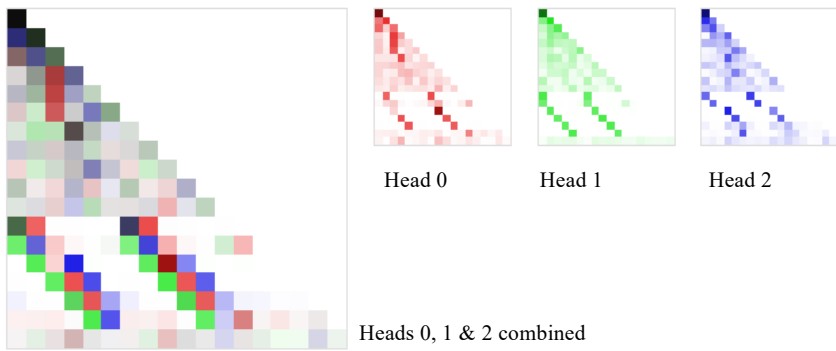

Figure 11: For 5 digit addition, using 3 attention heads gives the best separation with the heads having distinct, non-overlapping attention pattern. Ablating any head increases the model loss, showing all 3 heads are useful.

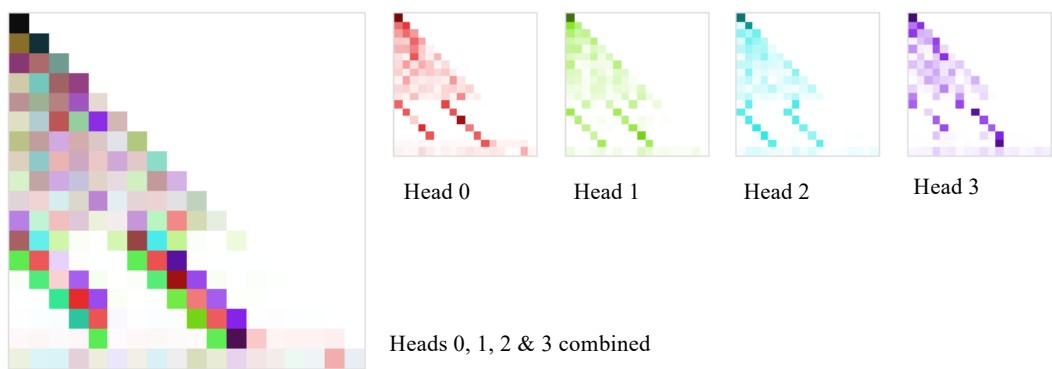

Figure 12: For 5 digit addition, using 4 attention heads gives an attention pattern where the H1 and H2 staircases overlap perfectly. Ablating one of H1 or H2 increases loss. The similarity in H1 and H2's attention patterns suggests they are "splitting" a single logical task. Spliting is feasible as Elhage et al. (2021) says attention heads are independent and additive.

## B  APPENDIX - TRAINING LOSS BY NUMBER OF DIGITS

The model can be successfully trained with 5, 10, 15, etc digits (Refer Figs. 13, 14, 15). For a given loss threshold, 15 digit addition takes longer to train than 10 digit addition, which takes longer to train than 5 digit addition.

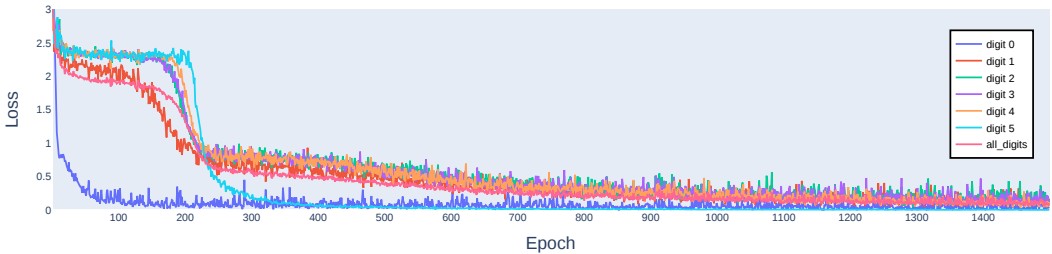

Figure 13: The per-digit loss curves over 1500 training epochs for **5** digit addition.

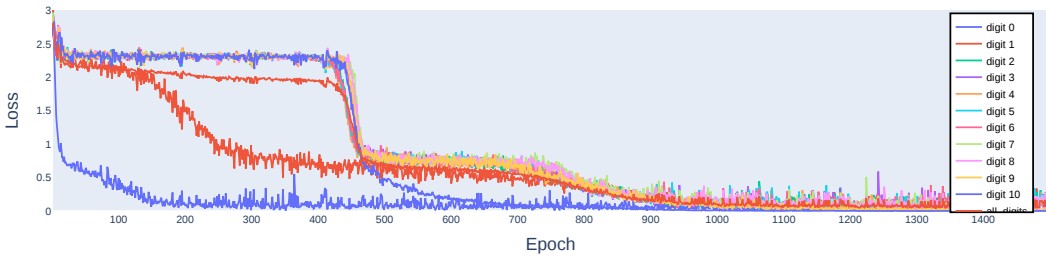

Figure 14: The per-digit loss curves over 1500 training epochs for **10** digit addition.

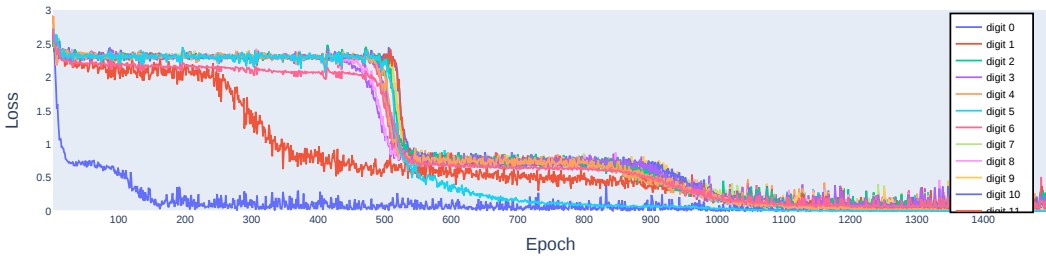

Figure 15: The per-digit loss curves over 1500 training epochs for **15** digit addition.

## C  APPENDIX - ATTENTION PATTERNS BY NUMBER OF DIGITS

The model can be successfully trained with 5, 10, 15, etc digits (Refer Figs. 16, 17, 18). The attention patterns for these cases show similarities that aid interpretation.

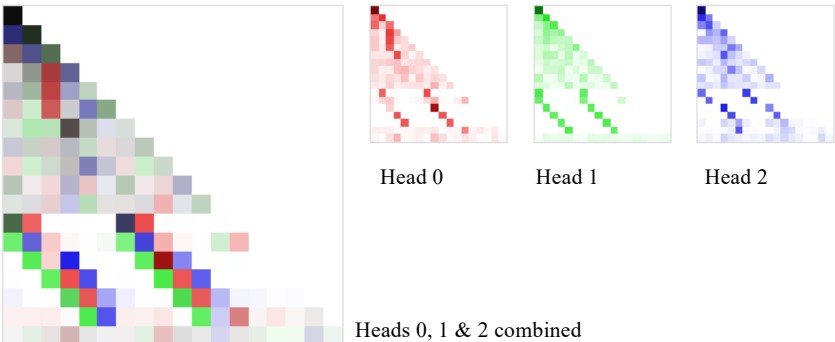

Figure 16: For **5** digit addition, and 3 attention heads, the attention pattern has a strong double-staircase shape. Each step of the staircase is 3 blocks wide, showing the attention heads are attending to different input tokens in each row.

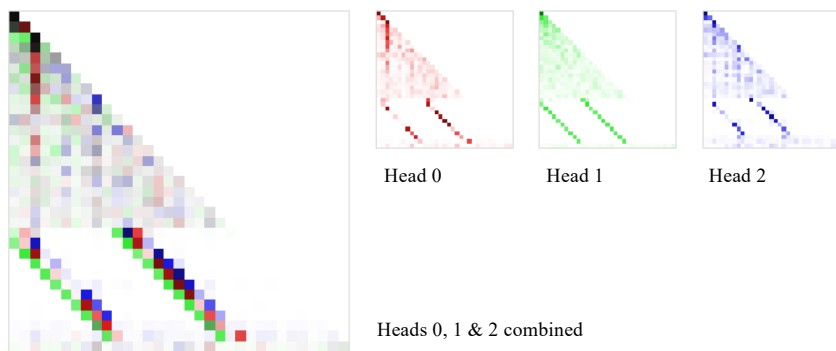

Figure 17: For **10** digit addition, and 3 attention heads, the attention pattern has a strong double-staircase shape. Each step of the staircase is 3 blocks wide, duplicating the Fig. 16 pattern.

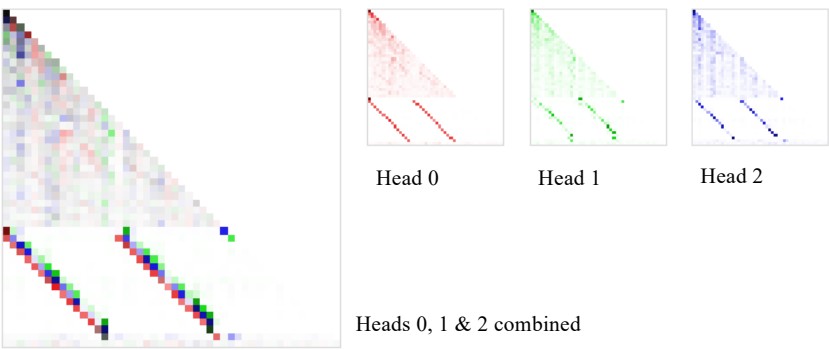

Figure 18: For **15** digit addition, and 3 attention heads, the attention pattern has a strong double-staircase shape. Each step of the staircase is 3 blocks wide, duplicating the Fig. 16 pattern.

## D    APPENDIX - MODEL CONFIGURATION

A Colab notebook was used for experimentation:

- It runs on a T4 GPU with each experiment taking a few minutes to run.

- The key parameters (which can all be altered) are:
  1. n_layers = 1; This is a one layer Transformer
  2. n_heads = 3; There are 3 attention heads
  3. n_digits = 5; Number of digits in the addition question
- It uses a new batch of data each training step (aka Infinite Training Data) to minimise memorisation.
- During a training run the model processes about 1.5 million training datums. For the 5 digit addition problem there are 100,000 squared (that is 10 billion) possible questions. So, the training data is much less than 1% of the possible questions.
- **US9** cascades (e.g. 44445+55555=100000, 54321+45679=1000000, 44450+55550=10000, 1234+8769=10003) are exceedingly rare. To speed up training, the data generator was enhanced to increase the frequency of these cases in the training data.

## E  APPENDIX - SOLVING A JIGSAW USING MANY PEOPLE

Understanding transformer algorithm implementation requires new ways of thinking.

Here is a useful analogy for changing how we approach a problem to more closely resemble how a Transformer would; a jigsaw puzzle. A single person could solve it using a combination of meta knowledge of the problem (placing edge pieces first), categorisation of resources (putting like-coloured pieces into piles), and an understanding of the expected outcome (looking at the picture on the box).

But if instead we had one person for each piece in the puzzle, who only knew their piece, and could only place it once another piece that it fit had already been placed, but couldn't talk to the other people, and did not know the expected overall picture, the strategy for solving the jigsaw changes dramatically.

When they start solving the jigsaw, the 4 people holding corner pieces place them. Then 8 people holding corner-adjacent edge pieces can place them. The process continues, until the last piece is placed near the middle of the jigsaw.

We posit that this approach parallels how transformer models work. There is no pre-agreed overall strategy or communication or co-ordination between people (circuits) - just some "rules of the game" to obey. The people think independently and take actions in parallel. The tasks are implicitly time ordered by the game rules.

## F  APPENDIX - USE SUM 9 TRAINING LOSS GRAPH

Fig. 19 shows a training loss curve for just **BA** and **UC1** tasks. The model has high loss on **Use Sum 9** cascades shown as high variability in the loss graph (See Fig. 20).

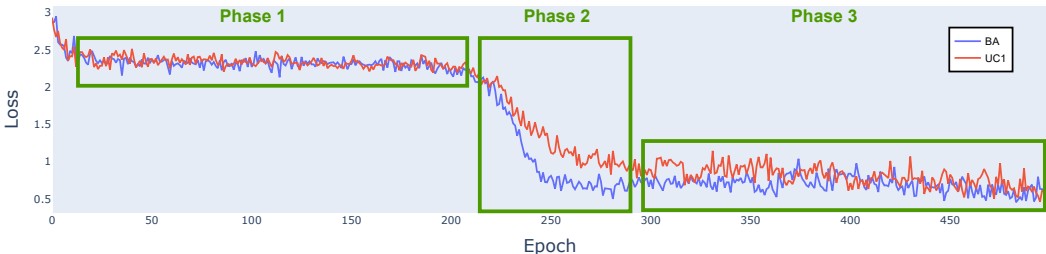

Figure 19: Training loss for digit 3 showing that, in Phase 2, the refining of **Use Carry 1** (red line) lags **Base Add** (blue line). This supports the claim that **Base Add** and **Use Carry 1** are learnt and refined separately and perform different calculation tasks.

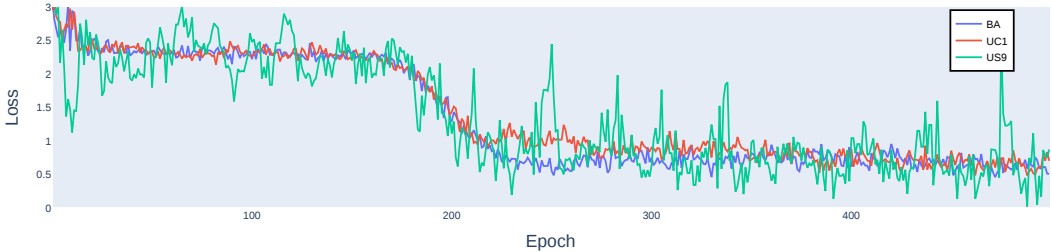

Figure 20: Training loss for digit 3 for the **Base Add, Use Carry 1** and **Use Sum 9** tasks, showing the model struggles to reduce loss on **Use Sum 9** (green line) compared to **Base Add** and **Use Carry 1** .

## G    APPENDIX - MODEL ALGORITHM AS PSEUDOCODE

The Algorithm 1 table below reproduces the model's addition algorithm summarised in Fig. 9 as pseudocode. This code calculates the largest digit first, handling simple (but not cascading) UseSum9 cases, and returns the answer.

Note that the pseudocode does not retain any information between passes through the **for** loop - this corresponds to each digit being calculated independently of the other digits.

## H    APPENDIX - THE LOSS FUNCTION AND LOSS MEASURES

The loss function is simple:

- Per Digit Loss: For "per digit" graphs and analysis, for a given answer digit, the loss used is negative log likelihood.
- All Digits Loss: For "all answer digits" graphs and analysis, the loss used is the mean of the "per digit" loss across all the answer digits.

The final training loss varies with the number of digits in the question as shown in Tab. 1. The final training loss for each digit in the question varies as shown in Tab. 2.

| Size of question | Final training loss | Example question |
|---|---|---|
| 5 digit addition | 0.009 | 11111 + 22222 = 033333 |
| 10 digit addition | 0.011 | 1111111111 + 2222222222 = 03333333333 |
| 15 digit addition | 0.031 | 111111111111111    +    222222222222222    =   0333333333333333 |
| **Overall Average** | **0.017** | **General Performance** |

Table 1: The final training loss after 5000 training epochs, each containing 64 questions, using All Digits Loss, for different size addition models.

We categorize each training question by which calculations its need to get the correct answer:

- **BA** : BaseAdd questions only need "base add" calculations.
- **MC1** : MakeCarry1 questions need both "use carry 1" and "base add" calculations.
- **US9** : UseSum9 questions need "use sum 9", "use carry 1" and "base add" calculations.

We measured the loss per question category as shown in Tab. 3. The frequency of these question types differ significantly ( **BA** = 61%, **MC1** =33%, **US9** =6% ) in the enriched training data so the final training loss values, at this level of detail, are not very informative.

After training, we used the model to give answers to questions. To understand whether the calculations at position n are important, we look at the impact on loss of ablating all attention heads at that token

---

**Algorithm 1** n-digit integer addition algorithm

---

1: **function** CALCULATEANSWERDIGITS($n\ digits, q1, q2$)
2:     answer $\leftarrow 0$                                           $\triangleright$ Initialize the answer to zero

3:     **for all** $i = 0, \ldots, n\ digits - 1$ **do**                         $\triangleright$ Loop over each digit
4:         pos $\leftarrow n\ digits - i - 1$                     $\triangleright$ Current position from the right
5:         prev pos $\leftarrow$ pos $- 1$                    $\triangleright$ Previous position from the right
6:         prev prev pos $\leftarrow$ pos $- 2$                $\triangleright$ Position before the previous

7:         mc1 prev prev $\leftarrow 0$           $\triangleright$ Start calculation of carry from two positions before
8:         **if** prev prev pos $\geq 0$ **then**
9:             **if** $q1_{\text{prev prev pos}} + q2_{\text{prev prev pos}} \geq 10$ **then**
10:                 mc1 prev prev $\leftarrow 1$          $\triangleright$ Carry from two positions before found
11:             **end if**
12:         **end if**

13:         mc1 prev $\leftarrow 0$              $\triangleright$ Start calculation of carry from previous position
14:         **if** prev pos $\geq 0$ **then**
15:             **if** $q1_{\text{prev pos}} + q2_{\text{prev pos}} \geq 10$ **then**
16:                 mc1 prev $\leftarrow 1$            $\triangleright$ Carry from the previous position found
17:             **end if**
18:         **end if**

19:         ms9 prev $\leftarrow 0$             $\triangleright$ Start calculation of sum of 9 in previous position
20:         **if** prev pos $\geq 0$ **then**
21:             **if** $q1_{\text{prev pos}} + q2_{\text{prev pos}} == 9$ **then**
22:                 ms9 prev $\leftarrow 1$              $\triangleright$ Sum of 9 in previous position found
23:             **end if**
24:         **end if**

25:         prev prev $\leftarrow 0$           $\triangleright$ Start calculation if carry from two positions before is needed
26:         **if** mc1 prev $== 0$ **then**
27:             **if** ms9 prev $== 1$ **then**
28:                 **if** mc1 prev prev $== 1$ **then**
29:                     prev prev $\leftarrow 1$          $\triangleright$ Carry from two positions before is needed
30:                 **end if**
31:             **end if**
32:         **end if**

33:         digitanswer $\leftarrow q1_{\text{pos}} + q2_{\text{pos}} +$ mc1 prev $+$ prev prev   $\triangleright$ Calculate answer for current digit
34:         digitanswer $\leftarrow$ MODULUS(digitanswer, 10)       $\triangleright$ Correct the current digit if it's $\geq 10$
35:         answer $\leftarrow$ digitanswer $+$ answer $\times 10$       $\triangleright$ Concatenate the current digit to the answer
36:     **end for**

37:     **return** answer                              $\triangleright$ Return the final calculated answer
38: **end function**

---

(using the TransformerLens framework, zero ablation of the blocks.0.hook_resid_post data set, the above loss function, and a "cut off" loss threshold of 0.08). Tab. 4 shows sample results. The results shows that all important calculations are completed at just 6 of the 18 (question and answer) token positions.

For deeper investigation, we created 100+ hand-curated test questions. They cover all question types (BA, MC1 and MS9) and all the answer digits (A5 .. A0) that these question types can occur in. These test cases were not used in model training. Example test questions include:

- make_a_question( _, _, 888, 11111, BASE_ADD_CASE)
- make_a_question( _, _, 35000, 35000, USE_CARRY_1_CASE)

| Digit index | Training loss | AKA | Example of digit |
|:---:|:---:|:---:|:---|
| 5 | <0.001 | A5 | 11111+22222=**0**33333 |
| 4 | <0.001 | A4 | 11111+22222=0**3**3333 |
| 3 | 0.003 | A3 | 11111+22222=03**3**333 |
| 2 | 0.008 | A2 | 11111+22222=033**3**33 |
| 1 | 0.046 | A1 | 11111+22222=0333**3**3 |
| 0 | 0.001 | A0 | 11111+22222=03333**3** |
| **Overall Average** | **0.010** | | |

Table 2: For 5-digit addition, final training loss per digit. The more digits, the longer the model needs to train to a given level of loss.

| Question type | Training loss | Aka | Example of digit |
|:---:|:---:|:---:|:---|
| **BA** | 0.021 | Base Add | 11111+22222=033333 |
| **MC1** | 0.001 | Make Carry 1 | 11811+22222=034033 |
| **US9** | <0.001 | Use Sum 9 | 17811+22222=040033 |
| **Overall Average** | **0.011** | **General Performance** | |

Table 3: For 5 digit addition, the final training loss per question category.

- make_a_question( _, _, 15020, 45091, USE_CARRY_1_CASE)

- make_a_question( _, _, 25, 79, SIMPLE_US9_CASE)

- make_a_question( _, _, 41127, 10880, SIMPLE_US9_CASE)

- make_a_question( _, _, 123, 877, CASCADE_US9_CASE)

- make_a_question( _, _, 81818, 18182, CASCADE_US9_CASE)

The above experiment was repeated (with "cut off" loss threshold of 0.1) using these ~100 questions (and another 64 random questions). We analyzed the incorrect answers, grouping these failures by which answer digits were wrong. Tab. 5 shows sample results.

| Row | Number of incorrect answers, grouped by incorrect digits |
|:---:|:---|
| 11 | **Nyyyyy: 47**, NyNyyy: 5, yNyyyy: 7, yyNyyy: 7, yyyNyy: 1, ... |
| 12 | **yNyyyy: 97**, yNNyyy: 7, NNyyyy: 3, Nyyyyy: 2, yyyNyy: 1, ... |
| 13 | **yyNyyy: 85**, NyNyyy: 3, yNNyyy: 2, Nyyyyy: 3, yNyyyy: 5, ... |
| 14 | **yyyNyy: 72**, yyNNyy: 4, NyyNyy: 3, yNyNyy: 3, Nyyyyy: 2, ... |
| 15 | **yyyyNy: 74**, yyNyNy: 3, Nyyyyy: 2, NyyyNy: 3, yNyyNy: 3, ... |
| 16 | **yyyyyN: 82**, yyNyyN: 2, NyyyyN: 4, yNyyyN: 4, yyNyyy: 9, ... |

Table 5: For 5 digit addition, ablating all the attention heads in each row (aka step) impacts answer correctness in a regular pattern. Here, an 'N' means that answer digit was incorrect in the predicted answer. In each step, the failure count for the top grouping is > 6 times the next most common grouping - evidence that each step calculates one answer digit. From row 11, each row mainly impacts one answer digit - in the same order as the model predicts answer digits.

We also ablated one head at a time (using the TransformerLens framework, mean ablation of the blocks.0.attn.hook_z data set, the standard loss function, and a "cut off" loss threshold of 0.08) to understand which head(s) were key in calculating BA questions. Tab. 6 shows sample results.

The last row of Tab. 6 shows that digit A0 was calculated in step 16. This is one step before the model reveals A0 in step 17. The other rows in the table shows that this pattern holds true for all the other digits: each digit is calculated one step before the model reveals it.

| Token | Average loss | Conclusion |
|-------|--------------|------------|
| 0 .. 10 | 0.070 | Impact is low. Calcs for these 11 tokens are unimportant |
| 11 | 0.322 | Loss is 4 x threshold. Calculations are important |
| 12 | 0.497 | Loss is 6 x threshold. Calculations are important |
| 13 | 0.604 | Loss is 7 x threshold. Calculations are important |
| 14 | 0.921 | Loss is 11 x threshold. Calculations are important |
| 15 | 1.181 | Loss is 14 x threshold. Calculations are important |
| 16 | 1.021 | Loss is 12 x threshold. Calculations are important |
| 17 | 0.070 | Impact is low. Calculations for this token is unimportant |

Table 4: For 5 digit addition, for the 18 tokens / calculation rows, how ablating each token/row impacts the calculation loss.

| Ablated Head | Average loss | Conclusion |
|--------------|--------------|------------|
| 0 | 0.016 | Impact is low. Head is unimportant for **BA** |
| 1 | 4.712 | Loss is 50 x threshold. Head is important for **BA** |
| 2 | 0.062 | Impact is low. Head is unimportant for **BA** |

Table 6: For 5 digit addition, when ablating heads, Head 1 is clearly key for the calculation of **Base Add** questions.

# I    APPENDIX - ALGORITHM RE-USE

We explored whether the algorithm is re-used in other models by training a separate 1-layer 5-digit addition model with a different seed, optimiser. It also required the model to predict "+" as the first answer token as shown in Fig. 3.

Figure 21: We refer to individual tokens in a 5-digit addition question as D4, .. D0, and D'4, .., D'0 and the answer tokens as A6, .., A0. Note that A6 is always the "+" token.

Figs. 24, 25 and 23 show aspects of the behavior of this model. Fig. 26 shows the nodes in the model where the node purpose can be identified by automated search using behavior filtering and ablation intervention testing.

| Posn | P0 | P1 | P2 | P3 | P4 | P5 | P6 | P7 | P8 | P9 | P10 | P11 | P12 | P13 | P14 | P15 | P16 | P17 | P18 |
|------|----|----|----|----|----|----|----|----|----|----|-----|-----|-----|-----|-----|-----|-----|-----|-----|
| Posn | D4 | D3 | D2 | D1 | D0 | + | D'4 | D'3 | D'2 | D'1 | D'0 | = | A6 | A5 | A4 | A3 | A2 | A1 | A0 |
| # fails | . | . | . | . | . | . | . | . | . | . | . | . | 467 | 804 | 765 | 775 | 724 | 582 | . |

Figure 22: This shows the token positions where questions fail when we ablate each node in the 5-digit 1-layer 3-head addition model. The model only uses nodes in token positions P12 to P17 (answer digits A6 to A1).

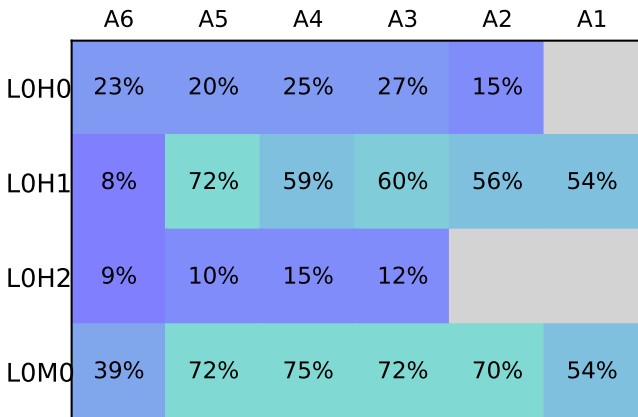

Figure 23: This map shows the **percentage of enriched questions** that fail when we ablate each node in the 5-digit 1-layer 3-head addition model. The model only uses nodes in token positions P12 to P7 (answer digits A6 to A1). Lower percentages correspond to rarer edge cases. The grey space represents nodes that are not used by the model.

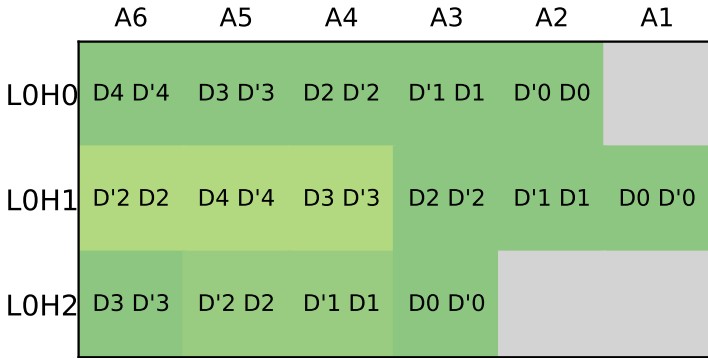

Figure 24: This map shows the input tokens each attention head attends to at each token position in the new model. The new model only uses attention heads in token positions P12 to P17 (answer tokens A6 to A1). Each head attends to a pair of input tokens. Each column attends to three different input token positions. These behaviors are the same in the new and previous models.

|  | A6 | A5 | A4 | A3 | A2 | A1 |
|---|---|---|---|---|---|---|
| L0H0 | A5 | A4 | A3 | A2 | A1 | |
| L0H1 | A5 | A4 | A3 | A2 | A1 | A0 |
| L0H2 | A5 | A4 | A3 | A2 | | |
| L0M0 | A5 | A4 | A3 | A2 | A1 | A0 |

Figure 25: This map shows the answer digit(s) A0 .. A5 impacted when we ablate each attention head and MLP layer in the new model. This behavior is the same in the new and previous models. Note that A6 is the constant "+" answer token.

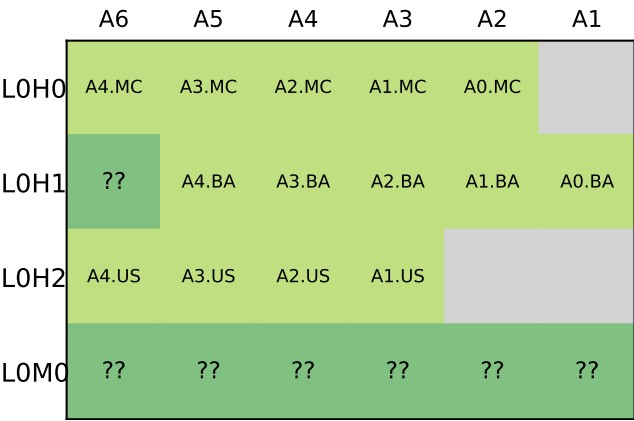

Figure 26: This map shows the calculation tasks **BA, MC1** and **US9** located in this model by automated search using behavior filtering and ablation intervention testing. For nodes containing "??" the automated search did not identify the purpose of the node.

