# OpenReview forum: "Understanding Addition in Transformers"
_ICLR.cc/2024/Conference — ICLR 2024 poster_

### Official Review · Reviewer_ExS5 · 2023-10-30

**Soundness:** 3 good
**Presentation:** 4 excellent
**Contribution:** 4 excellent
**Rating:** 8
**Confidence:** 3

**Summary:**

The paper studies the interpretability of Transformers. The authors focus on the 5-digit number addition task and analyze how a one-layer Transformer model finishes this task. To understand the model clearly, they propose a mathematical framework for integer addition, consisting of five tasks: Base Add, Make Carry 1, Make Sum 9, Use Carry 1, and Use Sum 9. The first three tasks can be independently executed for each digit pair, representing the sum of two digits modulo 10, checking for carry, and determining if the addition results in 9, respectively. The last two tasks chain operations across digits, respectively denoting adding the previous column's carry to the sum of the current digit pairs, and propagating a carry when Make Sum 9 and Use Carry 1 are true. The authors then analyze the one-layer Transformer model under this framework during the training phase and testing phase. More precisely, during the training phase, the authors investigate the training loss for Base Add (BA), Use Carry 1 (UC1), and Use Sum 9 (US9) three tasks. According to the experimental results,   US9 is the most complicated, especially in the case where more than one column carry occurs (e.g. 445+555=1000) and BA, UC1 two tasks are highly correlated. During the testing phase, the authors use ablation experiments to evaluate each attention head and conclude that for different digit pairs, the model uses slightly different algorithms.

**Strengths:**

- The paper advances in the direction of opening the black box of Transformers, which is a very important topic as Transformers are being applied in an increasing number of domains.

- The authors decompose integer addition into several subtasks and investigate the loss of each task during the training. This might provide some inspiration for future improvements in deep learning for math.

- The paper is well-organized. The basic idea is clean and easy to follow.

**Weaknesses:**

- One experimental flaw is that the test accuracy of the model is not provided. In addition, it is also worthwhile to explore using the trained model directly for the addition of integers with more digits.

- In the integer addition task, digit 0 should be treated as a special case, since intuitively, when humans perform integer calculations, the more zeros there are, the easier the calculation becomes. In other words, the digit 0 requires special attention. So it might be interesting to include Make Sum 0 and Use Sum 0 in the mathematical framework.

**Questions:**

(1) What's the performance of the trained model on the test data?

---

> ### Author Response · Authors · 2023-11-21
> **Response to reviewer ExS5's review**
>
> We thank the reviewer for the comments and feedback. They helped improve the paper.
>
> We now address the weaknesses and questions raised point by point:
>
> __Weakness: One experimental flaw is that the test accuracy of the model is not provided. In addition, it is also worthwhile to explore using the trained model directly for the addition of integers with more digits.__
>
> The new Appendix 17 provides this detail, including loss for 5, 10 and 15 digit cases.
>
> __Weakness: In the integer addition task, digit 0 should be treated as a special case, since intuitively, when humans perform integer calculations, the more zeros there are, the easier the calculation becomes. In other words, the digit 0 requires special attention. So it might be interesting to include Make Sum 0 and Use Sum 0 in the mathematical framework.__
>
> The Make Sum 0 ideas was indeed considered by the authors. Make Sum 0 is a useful efficiency gain for humans doing a specific sub-class of addition questions. Our intuition is that the model is calculating bigrams of tokens without any real understanding of numbers - it treats the digits as string tokens "3" + "4" = "7". The model can efficiently map all 100 pairings without needing to or benefiting from treating "0" as a special case. Our intuition is that if the model treated "0" as a special case, this would increase the algorithm complexity without increasing its accuracy and so the model would not benefit from adding this rule.
>
> __Q1. What's the performance of the trained model on the test data?__
>
> The overall response and the new Appendix 17 provide this detail.

---

> > ### Comment · Reviewer_ExS5 · 2023-11-22
> >
> > The reviewer thanks the author for their response. The new appendix strengthens the paper a lot in my opinion.

---

### Official Review · Reviewer_sWU9 · 2023-11-01

**Soundness:** 3 good
**Presentation:** 1 poor
**Contribution:** 2 fair
**Rating:** 3
**Confidence:** 4

**Summary:**

This paper delves into the intricacies of a one-layer Transformer model trained for integer addition, emphasizing the importance of understanding machine learning models for safety and ethical considerations. The study uncovers that the model breaks down the addition task into parallel, digit-specific streams, using different algorithms for various digit positions. Interestingly, the model starts its calculations later but completes them swiftly. A unique use case with a high loss is pinpointed and elaborated upon.

**Strengths:**

This is a technically solid, moderate to high impact paper, with no major concerns with respect to evaluation, resources, reproducibility, ethical considerations.

**Weaknesses:**

1. The adaptability of the methodology in this paper is limited, as it only applies to a one-layer Transformer model. Perhaps further analysis on two-layers or even more complex models would be beneficial. Moreover, the study solely focuses on integer addition, making it challenging to extend to other operations like subtraction or multiplication.
2. The writing of this paper is not comprehensive. For instance, the descriptions for Figure 4 and 8 are difficult to comprehend.
3. The experiments conducted are not exhaustive. In the "Prediction Analysis" section, the authors failed to provide a specific metric and results compared to baselines.

**Questions:**

Please see the Weaknesses.

---

> ### Author Response · Authors · 2023-11-21
> **Response to reviewer sWU9's review**
>
> We thank the reviewer for the comments and feedback. They helped improve the paper.
>
> We now address the weaknesses raised point by point:
>
> __Weakness: The adaptability of the methodology in this paper is limited, as it only applies to a one-layer Transformer model. Perhaps further analysis on two-layers or even more complex models would be beneficial. Moreover, the study solely focuses on integer addition, making it challenging to extend to other operations like subtraction or multiplication.__
>
> Feedback includes that analysis should be extended to two-layer or more complex models. We agree with this direction but see this work as outside the scope of this paper. When this paper was written we did not understand the more-complex 2-layer addition algorithm well. (Our analysis of this model is now well advanced. The 2-layer model utilizes the 1-layer algorithm but also includes an additional algorithm covering the “cascading US9” high-loss edge case. This will be the subject of a future paper.). We see this paper is a solid first step on a long journey explaining ever more complex algorithms (e.g. multiplication) partially by re-using well-understood foundational components (e.g. addition).
>
> The concern is raised that the paper’s methodology is not reusable and the applicability of the paper is limited. We respectfully disagree. Table 5 in the new Appendix 17 “The loss function and loss measures” shows one instance of this approach, showing the output from an automated test framework.
>
> The concern is raised that the methodology only applies to one-layer transformer model. While, it is very likely that 1-layer addition, 2-layer addition, subtraction, multiplication implement substantially different algorithms, this paper’s methodology provides a way to gain many insights into any algorithm, before the algorithm’s implementation detail is understood. While these insights do not explain the algorithm’s implementation directly, from experience they prove to be very useful in constraining the feasible algorithm space, leading to further researcher insights.
>
> __Weakness: The writing of this paper is not comprehensive. For instance, the descriptions for Figure 4 and 8 are difficult to comprehend.__
>
> The concern was raised that the paper was not comprehensive. In this paper we:
> - Provide an alternative mathematical framework for addition suited to neural networks
> - Show not only that the model can do addition and is low loss, but explains the edge-case causing the loss in terms of the mathematical framework
> - Propose an explanation for the model’s implementation of the addition algorithm in terms of the mathematical framework
> - Validate the proposed algorithm with experimental results. Details of this validation are in the new “Loss function and loss measures” appendix.
>
> A concern was raised that Figure 4 is difficult to understand. The addition algorithm that the model learnt is more complex than the traditional human process for addition. Understanding the model’s approach requires us to ignore our existing process, disassemble the addition process into its foundational components and then reassemble an alternative process. Figure 4, supported in the paper, documents this alternative approach. It is non-trivial to understand but led to insights key to understanding the model’s algorithm. We believe Figure 4 is useful and instructive.
>
> A concern was raised that Figure 8 is difficult to understand. The addition algorithm that the model learnt is complex - more complex than the traditional human approach to addition. Some complexity in description is unavoidable. Figure 1 introduces the algorithm concepts and shows a simplified representation of the algorithm. Figure 8 fleshes out the algorithm’s detail. The two figures work together, supported in the paper, to build out the concepts. We believe these figures are very good representations of the algorithm, and that further simplification in the diagram would require omitting pertinent information. To aid in comprehension, a new Appendix 16 “Model Algorithm as Pseudocode” reproduces the algorithm in Figure 8 as pseudocode.
>
> __Weakness: The experiments conducted are not exhaustive. In the "Prediction Analysis" section, the authors failed to provide a specific metric and results compared to baselines__
>
> The overall rebuttal and the new Appendix 17 “Loss function and loss measures” provide this detail.

---

> ### Author Response · Authors · 2023-11-22
> **Request for feedback**
>
> Dear Reviewer sWU9
>
> Thanks a lot for your time in reviewing and insightful comments, which we used to carefully revise the paper to answer your questions. We sincerely understand you’re busy. But since the discussion due is approaching, would you mind checking the response and revision to confirm whether you have any further questions?
>
> We are looking forward to your reply and are happy to answer your further questions.
>
> Best regards

---

### Official Review · Reviewer_f5y1 · 2023-11-02

**Soundness:** 3 good
**Presentation:** 3 good
**Contribution:** 3 good
**Rating:** 8
**Confidence:** 3

**Summary:**

Very interesting paper that focuses on explaining the "inner workings" of the foundational model of Transformer. While the use-case demonstrated (integer addition with a single layer transformer) is simplistic, the idea is novel and the visualisations are meaningful and make sense for better trust and confidence in how a transformer model works for the AI community.

**Strengths:**

Transformer model focus - no doubt an important model in the current AI landscape. Solid mathematical explainations and interpretation of the model working, the attention visualisations shown are very interesting and the model training loss curve which shows how a transformer trains individual digits semi-independently was promising to see.

**Weaknesses:**

No major weakness other than the paper applying the framework of explainability to a simple problem (integer addition). Though, this is well the strength as well of the paper as it makes the model easier to interpret and understand.

**Questions:**

Solid theoretical framework in the paper, good interpretation and visualisations - no further questions from this reviewer. The paper is very well written, easy to understand and the method is clear.

---

> ### Author Response · Authors · 2023-11-21
> **Response to reviewer f5y1's review**
>
> We thank the reviewer for the comments and feedback.

---

### Official Review · Reviewer_wEos · 2023-11-07

**Soundness:** 1 poor
**Presentation:** 2 fair
**Contribution:** 1 poor
**Rating:** 3
**Confidence:** 4

**Summary:**

This paper attempts to reveal the internal working mechanism of Transformers for integer addition tasks.

**Strengths:**

• Study an important problem of transformer models' application in numerical computation tasks.

**Weaknesses:**

• The analysis framework and analysis lacks mathematical rigidity.
	• The conclusion is not well established based on rigorous mathematical framework.
	• The paper does not fully utilize/choose the most relevant aspect of transformers for addition tasks.

**Questions:**

1. Page 2, a latex error: "d_e-dimensional embeddings"
	2. Page 3, section 3, paragraph 3, "detailing identified circuits", is this a typo? Or which "identified" circuits? What is the "identified" process?
	3. Page  3, section 3, paragraph 4, "techniques in works like symbolically … ", a grammar error?
	4. Page 3, section 3, paragraph 7, "Surveys like  overview techniques…", missing citations after "Surveys like"?
	5.  Page 4, section 4, paragraph 3, "Fig. 2 shows … semi-indendently…", what is the loss per digit is being plot in Figure 2?
	6. Page 4, section 4, paragraph 4, "Transformer models always process text from left to right…", this is not true. It is just an artifact of GPT-style attention masking. For example, we can do config the attention mask to enable full order attention over the two addends and generate the outputs in all kinds of order, e.g. from the tens digit to higher value digits, from the middle digit to two ends, and so on. We can also do non-autogression generation, e.g. incremental masking output generation.
	7. Page 4, figure 3 caption "..After the question is fully revealed (at layer 11)..", by "layer 11" do you mean the 11th row? To avoid ambiguity, it is better to number the attention matrix and refer to them the row or column number across the paper.  Also what are the sub-figures of 0.0, 0.1, 0.2? Different heads? What are the labels?
	8.   Page 4-5, section 5, please clarify whether the  "mathematical framework" is  for characterizing (grouping) addition data instances-digits only? Or is there a link to the loss? If so, please formulate the framework and what kind of mathematical hypotheses this framework can verify formally in mathematical terms?  Also please detail the loss on each digits formally. Also please detail the statistics of your training and valid datasets in terms of your classification of digits in your framework.
	9. Page 4-6, please detail how the loss is being average. Are they per digits or per digit average?
	10. Page 6, please introduce or define or describe phase 1, 2, 3 formally?
	11. Page 7, section 7, "During model prediction we overrode … the model memory (residual stream)…", please detail the approach formally? Are your conclusions/assertions based on checking the attention scores?  Please discuss explicitly with formal treatment. Otherwise, the plain English language analysis in Section 7 is difficult to follow and justify. Also formally define "independent of every other digit", "most impact on loss" and define it based on measure statistics during model inference time.

---

> ### Author Response · Authors · 2023-11-21
> **Response to reviewer wEos's review - Weaknesses [1/2]**
>
> We thank the reviewer for their comments and feedback.
>
> Our paper presents an in-depth analysis of a one-layer Transformer model trained for n-digit integer addition, showing the model divides the task into parallel, digit-specific streams and employs distinct algorithms for different digit positions. A rare use case with high loss is identified and explained. The model’s algorithm is explained, with findings validated through mathematical modeling and rigorous testing.
>
> We now address the weaknesses and questions raised point by point:
>
> __Weakness: The analysis framework and analysis lacks mathematical rigidity.__
>
> A concern was raised that the conclusion is not well established based on rigorous mathematical framework. Please refer to the overall rebuttal and the new Appendix 17 that provides detail on the experimental results that underpin the paper’s claims.
>
> __Weakness: The conclusion is not well established based on rigorous mathematical framework__
>
> A concern was raised that the analysis framework and analysis lacks mathematical rigidity. Please refer to the overall rebuttal and the new Appendix 17.
>
> __Weakness: The paper does not fully utilize/choose the most relevant aspect of transformers for addition tasks.__
>
> A concern is raised that the paper does not fully utilize/choose the most relevant aspect of transformers for addition tasks. We’re not clear on what this refers to. Assuming this refers to using a 1-layer rather than a 2-layer model, when this paper was written we did not understand the more-complex 2-layer addition algorithm well. Our analysis of the 2-layer addition model is now well advanced, and leverages some of the methodologies and findings of this paper. We see this paper is a solid first step on a long journey explaining ever more complex algorithms and transformers.
>
> see below Response to reviewer wEos's review - Questions [2/2] for responses to questions.

---

> ### Author Response · Authors · 2023-11-21
> **Response to reviewer wEos's review - Questions [2/2]**
>
> __Q1, Q2, Q3, Q4. Various small issues__
>
> Thank you for pointing out these issues. They have been resolved in the paper.
>
> __Q5. Page 4, section 4, paragraph 3, "Fig. 2 shows … semi-independently…", what is the loss per digit is being plot in Figure 2?__
>
> The per-digit loss is detailed in the new Appendix 17.
>
> __Q6. Page 4, section 4, paragraph 4, "Transformer models always process text from left to right…"__
>
> A concern was raised about the statement “Transformer models always process text from left to right”. This has been fixed in the paper to read: This autoregressive transformer model processes text from left to right.
>
> __Q7. Page 4, figure 3 caption "..After the question is fully revealed (at layer 11)..", by "layer 11" do you mean the 11th row?__
>
> A concern was raised about the inconsistent use of the term “layer” in explanations. We have fixed the paper to consistently use the terms: “layer” only for number of layers in the Transformer and “row” for each (vertical) step in the attention pattern.
>
> The meaning of the labels in the attention pattern figure was queried. The Figure image has been improved in the paper to clarify the meaning of the image labels as Layer and Head.
>
> __Q8. Page 4-5, section 5, please clarify whether the "mathematical framework" is for characterizing (grouping) addition data instances-digits only? Or is there a link to the loss?__
>
> The concern about the independence (or otherwise) of the mathematical framework and the loss calculations is covered by the “Weaknesses” section above and the new Appendix 17. The “Mathematical Framework” section now concludes with: *We use this mathematical framework solely for analysis to gain insights. The model training and all loss calculations are completely independent of this mathematical framework.*
>
> __Q9. Page 4-6, please detail how the loss is being average. Are they per digits or per digit average?__
>
> There is a loss function used with per-digit graphs, and a separate "mean" loss function used with all-digit graphs. Both loss functions are defined in the overall feedback and in the new Appendix 17.
>
> __Q10. Page 6, please introduce or define or describe phase 1, 2, 3 formally?__
>
> The suggestion is made to introduce or define the phases 1, 2 & 3 formally. We are hesitant to make strong claims about these phases at this time. Experimentally, and as per the literature, they seem to correspond to “memorization”, “algorithm refinement” and “clean-up” phases. The “Training Analysis” figure description now includes: The 3 phases seem to correspond to “memorization”, “algorithm discovery” and “clean-up”.
>
> __Q11.Page 7, section 7, "During model prediction we overrode … the model memory (residual stream)…", please detail the approach formally__
>
> A request for a formal treatment of the conclusion/assertions including how the model prediction was overridden was requested is covered in Appendix 17.
>
> A request was made for a formal definition of the independent calculation of each digit. The new Appendix 16 “Model Algorithm as Pseudocode” shows how the algorithm calculates each digit independently.

---

> ### Author Response · Authors · 2023-11-22
> **Request for feedback**
>
> Dear Reviewer wEos
>
> Thanks a lot for your time in reviewing and insightful comments, which we used to carefully revise the paper to answer your questions. We sincerely understand you’re busy. But since the discussion due is approaching, would you mind checking the response and revision to confirm where you have any further questions?
>
> We are looking forward to your reply and are happy to answer your further questions.
>
> Best regards

---

> > ### Comment · Reviewer_wEos · 2023-11-23
> >
> > Thank the authors for your efforts to address my questions. For addition or for handling digits using transformer, besides 1 layer or multi-layer design choices, there are more options in how the sequencs of digits are being embedded,  prococessed and generated. The conclusion in the paper on transformer in handling addition jumps to the conclusion too quick without exploring other possibilities. It might be misleading. So I would keep my original rating.

---

> > > ### Author Response · Authors · 2023-11-23
> > > **Clarification needed**
> > >
> > > Thank you for your comment. We are not sure if we understand the reviewer's comment fully. Could the reviewer elaborate on "there are more options in how the sequencs of digits are being embedded, prococessed and generate".
> > >
> > > We'd also appreciate it if the reviewer could suggest what is misleading about our conclusion and if they could provide some feedback on that.

---

### Author Response · Authors · 2023-11-21
**Response common to multiple reviewers**

We thank the reviewers for their very useful comments and feedback.

A repeated concern was that there was a lack of detail around how loss is calculated and the results.

A new Appendix 17 “The loss function and loss measures” in the paper contains the loss metric definition, the baseline training loss, and how we overrode the model memory in experiments. It contains 5 tables detailing each experiment’s variation from baseline that provided evidence for the paper’s claims. It includes this information:

- The “per digit” loss calculation is negative log likelihood, while the “all answer digits” loss is the mean across the per digit losses.
- The training loss (after 5000 training batches each containing 64 datums) is 0.009 for 5-digit addition, 0.011 for 10-digit addition and 0.031 for 15-digit addition.

The other repeated concern related to how the mathematical framework and the training loss calculations interact:

We use the mathematical framework solely for analysis to gain insights. The model training and all loss calculations are completely independent of this mathematical framework.

The mathematical framework is an investigative tool to investigate the model’s algorithm, by stating some characteristics of what any accurate addition algorithm must perform. The framework is not trying to state exactly how the model implements the algorithm. Experimentation showed that, for this model, the implementation is closely aligned to the mathematical framework.

The mathematical framework is used to categorize questions (at both all-digits and per-digit level) into sub-groups (BA, MC1, US9) for analysis purposes only e.g. training loss graphs broken down by sub-group, and the impact of ablating heads on each sub-group. This approach generated several insights including 1) strong alignment of one step to each answer digit 2) strong alignment of one head to the BA task 3) strong evidence that answer digits are calculated one step before they are revealed and 4) several steps play no useful role in the answer calculations.

---

### Meta-Review · Area_Chair_DC2X · 2023-12-13

**Metareview:**

The paper provides a mechanistic interpretability analysis of a one-layer Transformer on an integer addition task. A major finding is that the model divides the addition task into parallel, digit-specific streams and employs different algorithms for different digit positions. It is also found that the model starts calculations late but executes them quickly.

Overall, the reviewers found that the paper does a solid, comprehensive, and novel analysis of the studied problem. There is no real concern from the reviewers other than that the paper focuses on a simple model on a single problem. From the AC's viewpoint, this paper studies a complex problem and makes sufficient contributions.

**Justification For Why Not Higher Score:**

Despite being an interesting study, this paper has a limited scope.

**Justification For Why Not Lower Score:**

See meta-review.

---

### Decision · Program_Chairs · 2024-01-16

Accept (poster)